# "It Will Lead You to Make Better Decisions about Your Health"—A Focus Group and Survey Study on Women's Attitudes towards Risk-Based Breast Cancer Screening and Personalised Risk Assessments

Jonathan Jun Kit Liow [1,†], Zi Lin Lim [1,†], Tomiko Mei Ying Sim [1], Peh Joo Ho [1,2,3], Su-Ann Goh [3], Sheen Dian Choy [4], Ying Jia Chew [5,6], Benita Kiat-Tee Tan [7,8,9,10], Veronique Kiak Mien Tan [7,8,9,10], Mikael Hartman [2,3,6], Keri McCrickerd [4,11] and Jingmei Li [1,2,*]

1   Laboratory of Women's Health and Genetics, Genome Institute of Singapore, A*STAR, Singapore 138672, Singapore
2   Department of Surgery, Yong Loo Lin School of Medicine, National University of Singapore and National University Health System, Singapore 117597, Singapore
3   Saw Swee Hock School of Public Health, National University of Singapore and National University Health System, Singapore 117549, Singapore
4   Human Development, Singapore Institute for Clinical Sciences, A*STAR Research Entities, Singapore 117609, Singapore
5   Department of General Surgery, Ng Teng Fong General Hospital, Singapore 609606, Singapore
6   Department of Surgery, National University Hospital, Singapore 119074, Singapore
7   SingHealth Duke-NUS Breast Centre, Singapore 168582, Singapore
8   Division of Surgery and Surgical Oncology, National Cancer Centre Singapore, Singapore 169610, Singapore
9   Department of Breast Surgery, Singapore General Hospital, Singapore 168753, Singapore
10  Department of General Surgery, Sengkang General Hospital, Singapore 544886, Singapore
11  Department of Paediatrics, Yong Loo Lin School of Medicine, National University of Singapore, Singapore 117597, Singapore
*   Correspondence: lijm1@gis.a-star.edu.sg; Tel.: +65-6808-8312
†   These authors contributed equally to this work.

**Abstract:** Singapore launched a population-based organised mammography screening (MAM) programme in 2002. However, uptake is low. A better understanding of breast cancer (BC) risk factors has generated interest in shifting from a one-size-fits-all to a risk-based screening approach. However, public acceptability of the change is lacking. Focus group discussions (FGD) were conducted with 54 women (median age 37.5 years) with no BC history. Eight online sessions were transcribed, coded, and thematically analysed. Additionally, we surveyed 993 participants in a risk-based MAM study on how they felt in anticipation of receiving their risk profiles. Attitudes towards MAM (e.g., fear, low perceived risk) have remained unchanged for ~25 years. However, FGD participants reported that they would be more likely to attend routine mammography after having their BC risks assessed, despite uncertainty and concerns about risk-based screening. This insight was reinforced by the survey participants reporting more positive than negative feelings before receiving their risk reports. There is enthusiasm in knowing personal disease risk but concerns about the level of support for individuals learning they are at higher risk for breast cancer. Our results support the empowering of Singaporean women with personal health information to improve MAM uptake.

**Keywords:** precision health; breast cancer; mammography screening; risk-based screening; focus group discussion; genetic literacy; health consciousness; health behaviour; qualitative research

## 1. Introduction

Breast cancer is the most common form of cancer among females globally and evidence strongly suggests that early breast cancer detection can save lives [1–4]. Currently,

mammography screening is considered the most effective method for finding the disease at an early stage [5]. Women who have undergone mammography screenings have been reported to be as much as 41% less likely to die from breast cancer within the next ten years [1].

Singapore has one of the highest age-standardised breast cancer incidence rates in Asia [6]. With the intention of lowering breast cancer mortality in Singapore, the national breast cancer screening programme (BreastScreen Singapore-BSS) was introduced in 2002—the first in Asia [7]. Through tailored invitation letters and subsidies, the BSS programme targets female Singaporeans and Permanent Residents 50 years of age and older for biennial screening. In addition, the local Health Promotion Board and various cancer societies hold activities and campaigns regularly to increase breast cancer awareness and promote the uptake of mammography screening. Women between the ages of 40 and 49 are encouraged to be screened but are urged to consult their doctor about the advantages and restrictions of screening mammography for their age group. However, even with a nation-wide screening programme, over 60% of Singaporean females between the ages of 50 and 69 years do not attend regular mammography at recommended intervals [7,8]. Studies over the past 25 years showed little change in women's barriers and motivations to attend mammography screening, with recurring main themes such as fear, cost, inconvenience, and a perceived lack of risk [9,10].

Given the low mammography uptake despite a multitude of efforts to increase accessibility to screening, a more efficient way forward might be to focus mammography screening efforts on those most at risk of developing the disease [11,12]. "For whom does screening work?" thus becomes a pertinent public health question. As a result of a better understanding of breast cancer risk factors, risk-based screening, as opposed to age-based screening, is currently being explored in a number of countries including Singapore [13–16]. Personalised breast cancer risk assessments may consider each woman's unique genetic, environmental, and lifestyle variations. In Singapore, the BREATHE (BREAst screening Tailored for HEr) pilot study uses the Gail model (non-genetic) and breast cancer polygenic risk to determine a woman's individual breast cancer risk [17]. BREATHE "aims to assess acceptability of a comprehensive risk-based personalised breast screening in Singapore". Individuals who had a mammogram performed within the past year or are willing to undergo mammography will have their risk adjusted based on their mammographic density and recall status. Women estimated to be at above average risk of developing breast cancer and were not recalled by a radiologist (i.e., for abnormalities found on their mammogram) were advised to consult a breast specialist in the study.

Among the three types of risk factors (genetic, environmental, lifestyle), genetic information may be the least familiar to the public [18,19]. However, the literature shows that there is general interest in knowing genetic testing results. In the United States, Croyle and Lerman indicated that most members of the general Utah population have generally positive attitudes and perceptions toward genetic testing for cancer [20]. Lerman et al. proposed that the interest in genetic testing could be due to women wanting to feel reassured and to learn about their children's potential risks and even aid childbearing decisions [21]. In the UK, women were found to have a high level of acceptance to risk-stratified screening with personalised support measures [22]. Family experience with cancer was another key motivator in a Gill et al. study with women in Australia and those women without extensive family history information were more motivated to seek genetic testing to fill this informational gap [23]. Similarly, in Singapore, Sun et al. identified that some Singaporean women are interested in genetic testing due to its perceived benefits, a strong family history of cancer, and their own desire to create awareness for themselves and their family [24]. Another study by Wong et al. found that "women were generally receptive towards SNPs (single nucleotide polymorphisms) gene testing" for personalised risk-based screening [25].

The advantages of personalised risk assessments and genetic testing may not always outweigh the disadvantages. A significant proportion of women continued to overestimate

their risk after receiving their assessment results and may require specific reassurance from specialised medical professionals if they are offered less frequent screening [26]. Schwartz also advised that while self-identified high-risk individuals may be more motivated to undergo genetic testing, the costs for genetic counselling and testing may affect one's readiness to undergo testing [27]. Women who perceived fewer personal barriers or costs of testing were four times more likely to want the test [28]. In addition, as Sierra et al. has noted, although genetic testing and information may be made available to the population, acting on these results is still up to the individual [29].

As studies from other countries have shown, in addition to a reduction in costs, risk-based screening can prevent more breast cancer deaths while maintaining the same number of false positives results as age-based screening [30]. However, are the women of Singapore ready for risk-based breast cancer screening? This research study will explore among women without a personal history of breast cancer: (1) attitudes towards the current breast cancer screening program, (2) attitudes towards risk-based screening, and (3) feelings and emotions on the prospect of receiving a personalised breast cancer risk report.

## 2. Materials and Methods

The study had two components; focus group discussions and surveys.

### 2.1. Focus Group Discussions

Research design:

The research design and report followed the COREQ checklist [31]. Eight online focus groups were conducted in Singapore to explore the attitudes towards mammography amongst women with no personal history of breast cancer. This research was part of a larger study that examined the attitudes of various stakeholder groups toward the mammography screening procedure. Details on patient sampling and recruitment have been described elsewhere [32]. The current study focused only on *women with no history of breast cancer*. All participants gave their informed consent to take part and the research was approved by the A*STAR IRB (Reference number: 2021-077).

Participant sampling and recruitment:

Briefly, individuals over 21 years old with an internet connection were eligible to participate. The recruitment process involved sending out flyers to various non-profit organisations via social media channels and email. After signing up, participants were invited to an online session where they gave their verbal consent and were briefed on the study. A total of 351 participants signed up for the main study by 14 March 2022, of which 140 gave their verbal informed consent; 211 did not respond to our email for informed consent. Of the 140 who consented, 70 participants were *women with no history of breast cancer*. Fifty-four participants took part in eight scheduled focus group discussions while a further sixteen were not able to attend any focus group session. A consistent response pattern emerged and participants brought up no new information from the sixth session onwards, thus confirming that data saturation had been reached [33].

Discussion guide:

The research team formulated the focus group questions to guide the discussion. The team, which comprises researchers and clinicians with backgrounds in breast cancer research, life sciences and psychology, has a track record with various methods to predict breast cancer risk and finding solutions to increase breast cancer screening uptake. Each focus group discussion began by exploring the women's perceptions on barriers and motivations toward mammography screening, in order to obtain a general sense of how they felt towards health and breast screening in general. Thereafter, the discussion was focused towards obtaining their views on a personalised risk assessment to understand its potential impact on their mammography screening behaviours. The participants were shown samples of the BREATHE risk reports for below average, average and above average breast cancer risks. A sample of the questions used in the guide are included in Supplementary Table S1. The main topics for the discussions were:

1.  Perceived barriers to mammography attendance and adherence;
2.  Motivations for attending mammography;
3.  Views towards personalised risk-based reports and genetic testing.

Data collection:

The focus groups were conducted over a secured online video conferencing application. Four to eight participants were assigned at random to each focus group to achieve an optimal number per group for fruitful discussions to occur [34].

The eight focus group discussions were conducted between July 2021 and March 2022. The same focus group guide was used by a trained facilitator to moderate each focus group session, which lasted between one and two hours. The main facilitator was a Chinese female aged 24, with a degree in Public Health and Life Sciences and experience in leading in-depth interviews while the back-up facilitator was a Chinese male aged 26, with a degree in Psychology and Communication and prior experience in conducting focus groups. There was no prior relationship between the facilitators and the research participants. All eight focus groups were conducted in English. At the end of each session, participants were thanked for their participation and given an SGD\$20 e-voucher each. The discussions were video- and audio-recorded and transcribed verbatim after the sessions. Transcript summaries of the FGD were returned to the participants for feedback and clarity.

Data analysis:

The transcripts were imported into the QSR Nvivo software package (Version 1.6, QSR International), in which a coding scheme was developed inductively based on the main topics in the discussion guide and the first two transcripts. This coding scheme was then refined and expanded using the subsequent data collected. Thematic analysis was conducted after five FGDs where the research team reviewed and came to an agreement on the main themes which the codes were grouped under to best explain the opinions and views of the participants. Coding continued for a subsequent three FGDs until data saturation was reached, after which no further focus groups were conducted. ZL and JJKL were the facilitators of the focus group discussions. Coding was carried out by three independent coders (ZL, JJKL and SA). SA has experience conducting in-depth interviews and has a degree in Biological Sciences and a MSc in Public Health. Representative participant quotes relating to the codes were chosen to explain each theme.

### 2.2. Survey in Individuals Participating in Risk-Based Breast Cancer Screening

To further assess a woman's response to risk-based breast cancer screening, a survey capturing emotion-related responses in anticipation of receiving a risk profile was administered to BREATHE participants [17]. Recruitment for BREATHE started in October 2021, at two restructured hospitals (Ng Teng Fong General Hospital and National University Hospital) and two polyclinics (Bukit Batok Polyclinic and Choa Chu Kang Polyclinic). As of 31 March 2022, nine-hundred and ninety-four participants (aged 35–59 years) completed the survey, with one withdrawn due to anxiety (final analytical dataset of nine-hundred and ninety-three participants). Participants completing the survey were not the same participants who attended the focus groups.

Participant responses were captured via a secure online questionnaire platform during the first visit, i.e., prior to the individual knowing their risk level [17]. Participants were required to state the extent to which they agreed with the statement "I am feeling <x> to receive my breast cancer risk report" on a five-point scale (*Strongly Agree, Agree, Neither Agree nor Disagree, Disagree,* or *Strongly Disagree*). Eight feelings were assessed in total, including five negative (*Scared, Regretful, Anxious, Worried,* and *Stressed Out*), and three positive (*Excited, Confident,* and *Optimistic*). These feelings were included upon consensus of the BREATHE study team members in the design of the study recruitment experience survey (Supplementary File S1). Participants' ratings reflected their feelings in anticipation of receiving their risk reports in ~3 months.

## 3. Results

### 3.1. Focus Groups

#### 3.1.1. Participant Characteristics

Table 1 provides the participants' characteristics. The participants (54 women) had a median age of 37.5 years [interquartile range: 10 years]. The age difference range in the various focus groups was between 13 and 31 years.

**Table 1.** Focus group discussion participant characteristics (women without personal history of breast cancer, *n* = 54).

| Characteristic | Number (%) |
|---|---|
| **Age** | |
| 21–29 | 8 (22.6%) |
| 30–39 | 21 (37.7%) |
| 40–49 | 18 (33.9%) |
| >50 | 7 (5.6%) |
| **Ethnicity** | |
| Chinese | 47 (86.7%) |
| Indian | 2 (3.7%) |
| Malay | 1 (1.8%) |
| Others | 4 (7.5%) |

#### 3.1.2. Using Thematic Analysis on Attitudes towards Breast Cancer Screening and Personalised Risk Assessments

Our analysis revealed two main themes: attitudes towards breast screening and attitudes towards personalised risk assessment. Each theme and their respective sub-themes are described below while Supplementary Table S2 lists the same points with more quotes from the participants.

1.  Attitudes towards breast cancer screening

Two sub-themes were identified: extrinsic (environmental) factors and intrinsic (personal) factors. Both themes were further split into barriers and motivations to breast screening. **Extrinsic factors** are external environmental factors affecting whether someone goes for a mammography. Extrinsic barriers include the **cost of mammography** (" . . . cost is a problem, so I might think about going maybe every five years, not every year" [F21]), **busy schedules** ("...sometimes you know life gets in a way, it's not something that is at the forefront of my mind" [F4]), and the **availability of information** ("it is not being published or is not we are not well informed of how this should be actually done" [F24]). Extrinsic motivators include **company healthcare policies** ("I would expect that this is part of my insurance, health check-up package, I believe that the mammogram will be part of that package, so I will go for it then" [F48]) and **presence of reminders** ("recently received a letter from MOH regarding pap smear now, so I think such letters are good to remind people, because without it I also wouldn't know, would not go for it" [F37]). Supplementary Table S2 shows the list of extrinsic factors and quotes.

**Intrinsic factors** are internal personal factors that come from a person's feelings, attitudes, beliefs and genetic make-up. Intrinsic barriers include **feelings of pain** ("...you're sort of shoved into the small plastic thing . . . but yeah it's kind of painful" [F53]), **pessimism towards diagnosis** ("...don't want to check la, now nothing la. Check already then I know then it becomes very sad" [F40]) and **perceived risk** ("...because I don't feel anything and I have no family history" [F52]), Intrinsic motivations include **family history** ("my father side on genetically my aunt has breast cancer" [F2]) and **having a peace of mind** ("It's just kind of peace of mind to make sure that everything is okay" [F18]). Supplementary Table S3 shows the list of intrinsic factors and quotes.

2.  Attitudes towards personalised risk assessments

In response to the sample breast cancer risk reports, the participants gave feedback on the risk reports. They commented that they wanted more information on the risk level rather than it being just listed as "Above Average" or "Below Average" ("Then you have to explain why you say that my risk is lower, yeah this one explain how you rate the risk" [F25]). The report was also said to be quite lengthy and could include more graphics instead (" . . . less wordy and more pictures" [F23]). The recommendations should also be more specific and actionable ("The recommendation is not very useful, I don't drink alcohol I don't smoke, I don't know what else I can do").

Nineteen of the fifty-four FGD participants outwardly said they welcome a risk-based screening paradigm, and many said they would be inclined to attend regular mammography after receiving a personalised breast cancer risk profile. Six FGD participants explicitly expressed that they did not feel that risk-based screening was useful or necessary.

The sub-themes we identified were related to (1) living in fear, (2) self-fulfilling prophecies, (3) rationalising fear and adapting positively to the information, and (4) suggestions to overcome the hurdles of increasing the acceptability of risk-based breast cancer screening.

Sub-theme 1        Living in fear

Participants said they would be living in fear if the report said they were at high risk of developing breast cancer. Many would be psychologically affected and anxious, affecting their lifestyle and life plans.

> *F10: It's not everyone that can cope in the positive way of knowing the risk, say okay I'm changing my whole life. They could change their life but become depressed. They're so scared of anything, they don't want children, they don't want to pass it on, they don't want to [do things]. Yes it's a risk of one in 1000, but I'm not taking that chance.*

> *F10: you'll just keep thinking about it. And sometimes, from my experience in genetic testing, sometimes if you give such a report, they might not get cancer at all, but they will be scared for life.*

Sub-theme 2        A self-fulfilling prophecy

A further deterioration of one's life could occur if the person receiving the risk report feels resigned that the estimated risk level determines their life outcome. In receiving a high-risk result, some might feel helpless and not change their lifestyle.

> *F39: may become a bit of a self-fulfilling prophecy. I don't know like if you are showing that you're high risk, then maybe you will think like aiya I will confirm get it and you continue with your unhealthy lifestyle.*

On the other hand, those receiving a low-risk result might be less inclined to maintain regular screening and a healthy lifestyle.

> *F31: Oh I'll consider not doing screening already.*

> *F28: I agree with F31, if my risk is below average then I will be less hard working.*

Sub-theme 3        Rationalising fear and adapting positively to the information received

Despite the above challenges, most participants felt that they could rationalise the fears and use their risk profiles to their advantage. Participants look forward to possessing the increased amount of information on themselves from the risk reports as they can use it to make better health decisions. They accept that while those people at a high risk of contracting breast cancer may need to go for a more regular mammography, there are those at a low risk that can reduce their number of mammography visits, thereby spending less money on their healthcare. Many also feel that any outcome is beneficial since more regular mammography visits mean the people at high risk can catch any potential breast cancer at an early stage.

> *F17: The more you know, I think it will be better. At first you'll feel scared right but still have to do something about it, for the sake of your children.*

> *F18: Yeah same thing. I mean at the most you'll be a bit more anxious. But I think it will lead you to make better decisions about your health as well.*

*F10: It's, this is a strange irrational thought, but it's really like you're looking for trouble. But of course, the rational side of you is saying that ... like you said, the sooner you know, the better, although your life would change to turn upside down, but at least, there is a better chance of fighting it.*

Sub-theme 4    Suggestions to improve acceptability of risk-based breast cancer screening

There are three ways that would help participants address their fears and uncertainty: a. clear communication of risk results, b. follow-up plans for personalised risk assessment, and c. providing actionable steps to manage their personal risk.

a.      Clear communication of risk results

The process of administering such tests must be properly carried out in such a way that the person fully understands the potential outcomes and consequences before performing the test.

*F19: I think it was mentioned in the beginning, that fear might be a factor in getting screening or getting your results or something. So, if fear is a factor, knowing more, if the information is not good, then it might deter some people.*

*F4: But I can also see how it will be confusing for people. Well, I think it's quite dangerous if it (the result) is misinterpreted, because ... individually everyone's interpretation of 70% risk would be very different.*

*F1: So, this means in the result, if you want to show high risk and low risk, then need to interpret. Like what does it mean low risk, what does it mean high risk.*

b.      Follow-up plans for personalised risk assessment

Despite a person fully understanding their risk level, participants expressed the need for follow-on support to help them come to terms with their results or to express their feelings and reactions. Calls for psychological help and support groups were raised by the participants for those who would be categorised as having a high breast cancer risk, while those who are at a lower risk should be encouraged to continue going for screening.

*F10: I'm concerned with those who will get 90% results, or prediction. Like you just drop the bomb then bye bye ... There should be some kind of plan all around. Like encourage those that are low risk to still go screening and those that are high risk, support them through their anxiety. Because you could be predicted to have 90% of getting it but that you never get, and then you live the rest of your life in fear.*

*F10: You know, so it can change your life, whether it's positive right and the person who gave you that information or did all those tests, they are no longer there. So if there is no comprehensive, discussion program, or psychologist and sociologist, then I'm not really sure [about it].*

*F24: If it's high risk, so I hope that there is a number, I can call in and seek for advice, what is the next step to be done ... So, if only they have a support group. Once if 'you're diagnosed is the above average not necessarily you contract, but that that is that you can actually call the support group where they can actually advice and encourage you to go for further checkup...*

c.      Providing actionable steps

A common view among participants was that the risk level meant little to them if they could not do anything about it. They wanted actionable steps that could be tailored to their lifestyle in order to improve their well-being and lower the risk of cancer.

*F25: Unless the doctor has a better way to prevent it, you know by letting us know, and they have a remedy to take for us, otherwise there's no point to let us know, it is better we don't know than to know.*

*F23: So what if we know about that risk, we cannot change anything so, what do you want us to do. I mean it is information and then so what? Then, that you go back to the*

*point that. People may not want to know at all, maybe they do not want to go testing because they cannot do anything about that you can achieve this cannot. Certain risk factors can like lifestyle smoking drinking, you can change them, but not your genes, So what can you do?*

*F46: I think actionable is very important. Like what F43 said. If there is nothing I can do about it, then don't tell me. But if you tell me you are at this risk then you do this this this to mitigate, or reduce your risk, then I think it's useful information.*

Sub-theme 5          Personalised risk assessment affects decision to go for screening
Despite all the above concerns, the final point is that the majority of participants feel that the personalised risk assessment is useful and will be more inclined to screen regularly after receiving a high-risk result.

*F4: I think all things considered, it probably will (make me screen regularly).*

*F7: I think this will definitely help. It will scare us but also help in our future screening. And we will keep this in mind, that we will need to get screening regularly, if there is a percentage.*

*F15: I agree. If I'm told I'm above average risk, I would be more strict about going for regular screenings.*

*F18: So if it's like an above average risk, it will definitely make me talk to my health care provider more about what I can do to help to make decisions about my health that can prevent me from getting breast cancer and also, I will definitely be more stringent about doing breast screenings.*

### 3.2. Survey of Individuals Participating in Risk-Based Breast Cancer Screening

As of 31 March 2022, a total of 993 women without a diagnosis of breast cancer completed the feedback survey. The median age at recruitment was 49 years (IQR: 43 to 54) (Table 2). The majority were Chinese (*n* = 764, 77%), followed by 119 (12%) Malay, 60 (6%) Indian, and 50 (5%) Others (Table 2). Approximately one in ten women reported positive family history of breast cancer (Table 2).

**Table 2.** Characteristics of 993 survey participants.

| Characteristics | Number (%) |
|---|---|
| **Median age** | 49 years |
| **Ethnicity** | |
| Chinese | 764 (77%) |
| Malay | 119 (12%) |
| Indian | 60 (6%) |
| Others | 50 (5%) |
| **Family history of breast cancer** | 108 (11%) |
| Mother | 50 (5%) |
| Sister | 60 (6%) |
| Daughter | 0 (0%) |
| **Ever attended breast cancer screening** | |
| Yes | 755 (76%) |
| No | 238 (24%) |

The majority of participants (*n* = 804, 81%) expressed positive emotions (i.e., strongly agree or agree) in anticipation of receiving their risk assessment reports, of whom 85% (688/804) expressed excitement (Figure 1). A small proportion (*n* = 374, 38%) expressed negative emotions, the most common was anxiety (*n* = 269, 72% of 374) (Figure 2). It should be noted that even though the participants felt anxiety, 76% (*n* = 204) expressed excitement. Figures 3 and 4 illustrate the number participants reporting each of the positive and negative emotions, respectively.

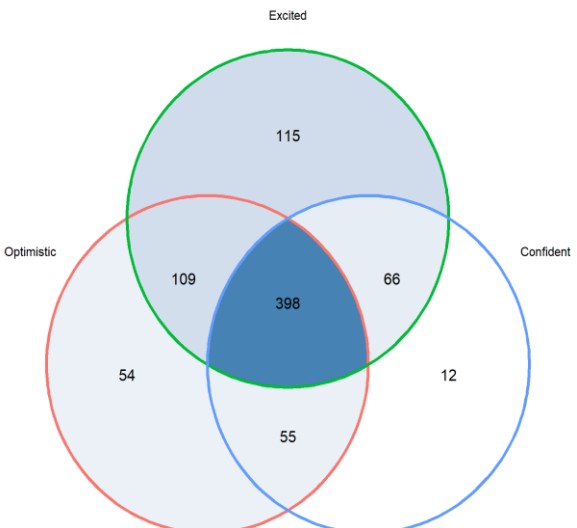

**Figure 1.** Positive emotions expressed (i.e., strongly agree or agree) by 809 of the 993 participants in the BREATHE study recruited between October 2021 and March 2022.

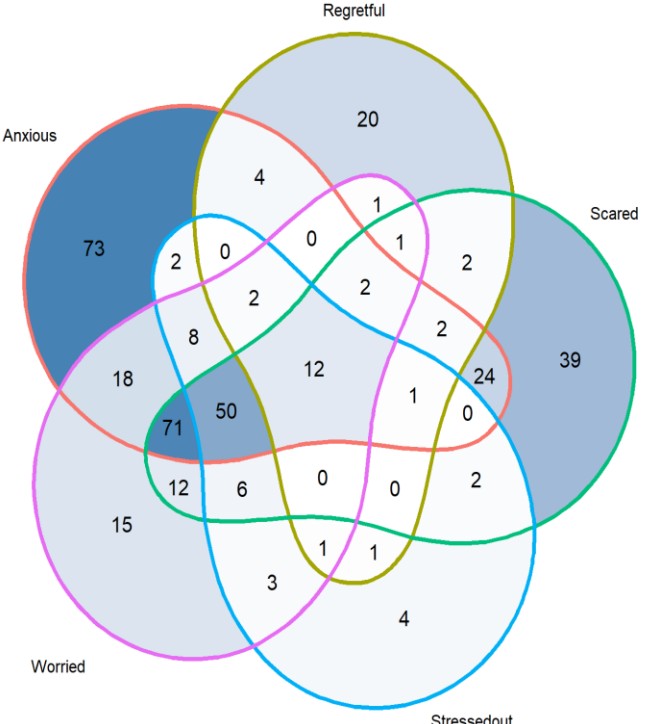

**Figure 2.** Negative emotions expressed (i.e., strongly agree or agree) by 374 of the 993 participants in the BREATHE study recruited between October 2021 and March 2022.

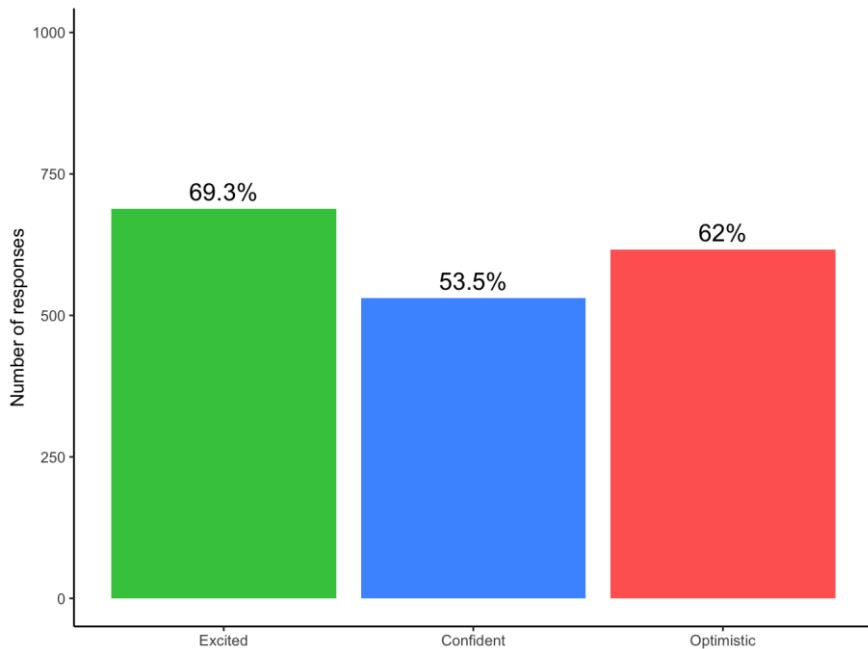

**Figure 3.** Barplot representing number of responses indicating each positive emotion among 993 participants.

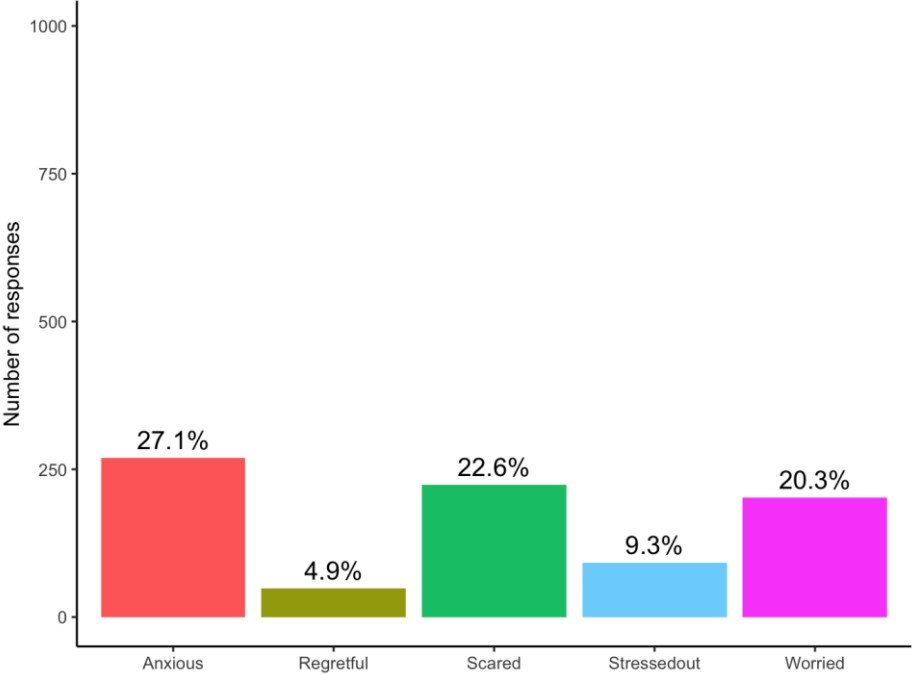

**Figure 4.** Barplot representing number of responses indicating each negative emotion among 993 participants.

## 4. Discussion

From our study we identified two main themes: attitudes towards mammography screening and attitudes towards personalised risk assessments. The barriers and motivations associated with using the mammography screening services offered in Singapore in this focus group study are largely similar to those reported in previous work. Under the second theme, participants might be living in fear if they received a high-risk classification, or feel helpless, leading to self-fulfilling prophecies. However, the participants said they were largely able to rationalise their fear and adapt positively to the new information and,

with further improvements to the risk assessments, felt that the reports would be useful. This finding was supported by results from our survey where women participating in a pilot risk-based screening study felt more positive emotions than negative emotions about the prospect of receiving their personalised breast cancer risk reports.

### 4.1. Attitudes towards Current Mammography Screening Programme Remained Largely Unchanged since the 1990s

Singapore is the first country in Asia to launch a widespread, population-based organised mammography screening programme in 2002 [6]. However, the consistently low participation rate (<40%) is far from the ideal minimum of 70% required to see substantial mortality reduction benefits [7,8,35]. Our eight focus group discussions eliciting opinions from 54 women without a history of breast cancer revealed no novel barriers and motivations to attending screening mammography in Singapore. Misconceptions about screening, lack of perceived breast cancer risk, perceived efficacy of mammography to save lives, physical discomfort, cost, inconvenience, fatalism, fear, support and encouragement from family members are reiterated considerations [9,35–39].

### 4.2. Current Strategies to Increase Breast Cancer Screening Uptake

A review examining strategies to encourage more women to take part in community breast cancer screenings in 14 studies found that promotional methods such as mailed educational material and phone calls increased mammography participation while methods such as letter of invitations, and home visits were no different to doing nothing at all in improving mammography attendance [40]. In another work, Yabroff et al. reported that these behavioural strategies (i.e., "strategies that alter cues or stimuli associated with screening behaviour", such as mail or phone reminders to screen) improved screening by 13.2% compared with no intervention in a meta-analysis that included 43 studies in the United States [41]. Analysing self-reported questionnaire data on mammography screening attendance and attitudes from 3739 breast cancer patients in Singapore, Lim et al. revealed various patterns among screeners who have recently undergone mammography screening [35]. The proportions of screeners who reported being motivated by both innate health consciousness and extrinsic cues, solely by innate health consciousness, solely by extrinsic cues, or a combination of other factors was found to be even (~25% in each group). The results suggest the use of a variety of different approaches to increase screening uptake. Many of these initiatives to increase screening uptake will likely have some impact. However, it is challenging to quantify the results from such initiatives to increase screening uptake.

### 4.3. Decision to Screen May Change for a Risk-Based Approach

Our study explored how a paradigm shift from the one-size-fits all approach to personalised feedback and recommendations may be received by the population targeted by mammography screening programs, i.e., women with no history of breast cancer. Our results suggest that participants mostly felt ready to use the information in their breast cancer risk level report in the future, but the risk information will only be useful if it is communicated properly and followed-up with appropriate support. As there are already extrinsic cues to encourage mammography screening, a multifactorial approach which includes raising intrinsic health consciousness may be an alternative way to address low participation rates [42]. Most countries use an age-based population-level breast cancer screening strategy that reduces breast cancer mortality but does not account for the wide variation in individual women's cancer risks [43,44]. A personalised screening strategy, where screening and prevention are based on a woman's risk of developing breast cancer, is possible thanks to increased knowledge of breast cancer risk factors [45]. A risk-based approach can potentially enhance the benefit-harm ratio of breast cancer screening programs [45]. Women are anticipated to gain personal health insights from tailored risk assessments, giving them the ability to work with their healthcare practitioners to make informed screening and prevention decisions [45]. Our results suggest that risk stratifica-

tion is a candidate mechanism to increase intrinsic health consciousness and/or intrinsic motivation to increase breast cancer screening rates. However, more work would be needed to support this claim.

Other studies have found similar positive attitudes, largely down to the women wanting more information about their health to make informed decisions. A risk-based approach, in particular, empowers women to evaluate their perceived susceptibility to the disease [42]. A study by Ghanouni et al. used face-to-face computer-assisted interviews on 933 women and found 85% think breast cancer risk assessments are a good idea [46]. An online survey by Bienge et al. on 4293 women aged from 30 to 69 years from Canada found 63.5% to 72.8% of them to be favourable towards risk-stratified breast cancer screening [47]. Nevertheless, results may differ between studies according to factors including age, education level, marital status, socioeconomic status, perceived risk, history of breast cancer, prior screening, and history of genetic testing for the disease.

One important aspect of breast cancer risk prediction is genetics. Our focus groups found participants to be generally keen on the introduction of risk-based breast cancer screening that includes genetic tests. Genomic technology has quickly proliferated around the world as a result of precision medicine programs, prompting concerns about genetic privacy and the ethics of data sharing [48]. It has been shown via previous research in bioethics and science and technology studies that different countries have distinct expectations regarding trust, openness, and public reason in connection to developing technologies and their administration [48]. It is thus critical to recognise the diversity of our Singaporean population in terms of acculturation, education, health knowledge, and cultural values to provide culturally sensitive and appropriate care. In Singapore, it was revealed in an online public survey ($n = 560$) that there is broad support for the use of genetic tests [48]. The actual experience of going through a genetic screen alleviated the fear of genomic technologies. However, the participants voiced persistent concerns about unprotected data sharing and a desire for ownership over their genetic information. The authors concluded that genetic education and exposure to genetic testing will help to garner support for genomic projects, precision medicine, and biobanking activities. Another study by Cheung et al. showed that the Singaporean population is moderately genetically literate but there are widespread privacy concerns over data sharing and regulation [48]. Nonetheless, their data demonstrates broad acceptance for the use of genetic testing in Singapore [48]. However, there remain valid concerns that should be considered when deciding how to deliver the risk report and follow-up.

As this risk-based screening is a relatively new concept to the public, there is uncertainty both in how accurate the results are and a lack of clear understanding on how the results would impact screening behaviour and lives [49]. The concept of flagging a healthy individual in screening programs, especially using genetic data, as high-risk for developing a complex disease is not easy to grasp. Differing from a medical diagnosis, where it is clear a person has a condition that can be clinically managed or treated, high-risk individuals have not developed the condition yet, and may never do. Our study revealed that receiving news of being at high risk of breast cancer may catch people off guard—some women felt that it may be better for them not to know their personal risk. Knowing one is at high risk but not knowing *when* or *if* breast cancer will develop may result in negative emotions associated with fear, powerlessness, and the belief that there is no escaping the disease, which may negate the benefits of risk-stratification [29]. Discussing breast cancer risk with at-risk women and promoting risk reduction techniques may thus cause anxiety and breast cancer worry [50–52]. Mental and emotional distress may result if women overestimate their risk of developing the disease [53,54]. However, the women in our focus group discussions also pointed out that knowing that they are at high risk would also motivate them to find out more from their doctors and be more disciplined at attending regular screening. Likewise, Evans et al. reported that in comparison to women not classified as high risk, women at high risk for breast cancer were substantially more likely to acknowledge their breast cancer risk is high and to adhere to routine screening [55]. Results from the Personal

Breast Cancer (BC) Risk Assessments (PBCRA) study (*n* = 31, semi-structured focus group discussions or interviews) showed that the majority of women appreciate the chance for a risk-based screening programme to guide enhanced disease monitoring, but they were reluctant in accepting decreased surveillance if estimated to be at low risk since they are comforted by routine screening [29,56]. This is at odds with our focus group results where most participants, when told to consider a scenario where they are classified as "low risk" of developing breast cancer, would feel complacent and reduce their mammography visits or stop them entirely. Others said they would feel motivated or relieved about the low risk as they can maintain their current lifestyle.

If there is just as much variability in response to risk results as there is variability in individual risk, effective communication of risk, recommendations, and consequences are essential if risk-based screening were to be implemented [22]. The healthy women in our focus group study highlighted a need for the proper communication of the risk results and additional follow-up plans, and that sufficient support should be provided for women who are classified to be at high risk for breast cancer. It was also brought up that those who are at a lower risk should be encouraged to continue screening, albeit less frequently. Furthermore, participants have raised concerns on their insurance coverage being affected as a result of a high-risk classification. As this new information comes to light, participants are unsure if they will need to disclose the results and potentially be excluded from medical plans covering breast cancer or pay a higher premium for the same coverage. Personal breast cancer risk assessments for healthy women may thus benefit from the development of education material, similar guidelines and clear recommendations that already exist for hereditary cancer [49,57–61]. As of present, the findings of predictive genetic tests cannot be used by Singaporean life insurers to evaluate or determine the outcome of insurance applications [62].

### 4.4. Women Attending a Pilot Risk-Based Screening Programme Look Forward to Receiving Personalised Breast Cancer Risk Assessment Results

To gain perspectives of a representative sample of a risk-based breast cancer screening population in Singapore, we looked at the survey responses from women who joined the BREATHE study. The quantitative results from the survey support the insights obtained from the focus group discussions. Respondents felt more positive than negative emotions towards the idea of receiving breast cancer risk assessment results. "Excited" was the most common feeling. More than a third of the respondents strongly agreed that they felt all three emotions including "Excited", "Optimistic" and "Confident". The high level of optimism towards risk-based breast cancer screening has also been reported by other studies. For example, in Kelley-Jones et al.'s study, women in the United Kingdom responded favourably towards personal breast cancer risk assessment and found it desirable to accept feedback on their individual risk [22]. However, it should be noted that our survey was administered to women who chose to participate in a pilot risk-based screening programme for breast cancer. These women may be inherently more health conscious and enthusiastic about learning their own risks. In addition, it has been reported previously that women's views toward breast screening may not be significantly changed by the concept of risk-based screening if they are already sceptical about screening in the first place (i.e., non-screeners) [22].

### 4.5. Caveat: Is the Public Understanding of Genetics Clear?

The public's grasp of genetics is widely believed to be inadequate. A particular issue to address is who should understand what and for what objectives. Experts must be thoughtful about the boundaries of their own expertise as well as the diverse demands of the various potential consumers of genetic knowledge in order to improve public understanding. In recent years, there has been a lot of media interest that oversells the potential of genetic discoveries, but this coverage is not always helpful and can even be deceptive to families who are at risk for genetic disease [63]. What exactly constitutes

genetic knowledge in the existing literature is ambiguous. In a healthcare setting, a health professional orders genetic tests to see if an individual has inherited a genetic mutation, and that person interprets and communicates the results to the patient. Direct-to-consumer (DTC) genetic testing, on the other hand, allows people to buy genetic tests and receive results without the need for a doctor's help [64]. Risk-based breast cancer screening studies, such as BREATHE, predicts breast cancer risk based on both high-risk breast cancer genes (ATM, BRCA1, BRCA2, CHEK2, PALB2, BARD1, RAD51C, RAD51D, or TP53) and common-variant-based breast cancer polygenic risk scores [17,65,66]. While our work revealed general optimism towards risk-based breast cancer screening, further work is needed to identify whether support is different for different risk tools.

Finally, it is also important to note a number of other considerations that limit the findings of our study. As the participants in the FGDs were recruited via online platforms, the participants tended to be technology-savvy people. As women with a higher household income are more likely to screen for breast cancer, it is likely that risk-based assessments will not encourage mammography screening attendance amongst women with lower incomes [67]. In addition, the median age of the FGD participants is below 50, the age at which women are invited by the national screening programme in Singapore to attend screening. The education material that comes with the screening invitation may influence the disease awareness of the women. This study population is on women without breast cancer only. Women with a previous history of breast cancer might have differing views and understanding of the risk reports based on their experiences. The results would also likely differ for groups with different social and economic backgrounds and challenges.

### 4.6. Recommendations

We have several suggestions to improve personalised risk assessments based on the participants' feedback. Firstly, by increasing the involvement of women in the design of risk reports, they are more likely to buy-in to the idea of personalised risk assessments and a risk-based breast cancer screening programme. With their involvement, the information in the report would also be more relatable and comprehensible to recipients of the risk report. There is an increasing emphasis on patient and public involvement (PPI), which seeks to actively engage participants in the design of research studies, policymaking, or programme development [68]. Several studies have included (PPI) to improve aspects of healthcare research such as care pathways [69], health technology assessments [70] and treat-to-target approaches [71]. We also recommend that in countries such as Singapore where healthcare is not universal, insurance information on whether one's risk levels will affect their insurance coverage be easily available so that the public can be reassured that they will not be negatively impacted by a high risk level.

## 5. Conclusions

A different screening approach may be needed to address the low mammography screening participation rates in Singapore. A risk-based approach is promising—high, but not universal, acceptability, was observed. However, it is anticipated that implementing it as a population-wide screening would present challenging problems. Proper communication of risk results and follow-up plans are needed to ease fears and guide women to make informed decisions on their health. Nonetheless, our study participants generally welcomed the adoption of risk-based breast cancer screening despite the obstacles and restrictions found. Our results from FGD and survey responses support the empowering of women with personal health information to increase health ownership and consciousness to improve screening uptake and adherence.

**Supplementary Materials:** The following supporting information can be downloaded at: https://www.mdpi.com/article/10.3390/curroncol29120719/s1, Table S1: Question Guide; Table S2: Extrinsic (environmental) factors and quotes on barriers and motivations to mammography; Table S3: Intrinsic (personal) factors and quotes on barriers and motivations to mammography.

**Author Contributions:** Conceptualization, J.L. and K.M.; methodology, J.J.K.L., Z.L.L., P.J.H., S.D.C. and K.M.; transcripts, J.J.K.L., Z.L.L., T.M.Y.S. and S.-A.G.; formal analysis, J.J.K.L. and Z.L.L.; writing—original draft preparation, J.J.K.L., T.M.Y.S. and J.L.; writing—review and editing, J.J.K.L., Z.L.L., T.M.Y.S., P.J.H., S.-A.G., S.D.C., B.K.-T.T., V.K.M.T. and M.H.; survey data provision, Y.J.C. and M.H.; supervision, J.L. and K.M.; project administration, J.J.K.L.; funding acquisition, J.L. and K.M. All authors have read and agreed to the published version of the manuscript.

**Funding:** This study is supported by the Agency for Science, Technology and Research (A*STAR) under its Social Sciences Innovation Seed Fund (C211618001). BREATHE is funded by the JurongHealth Fund (reference number JHF-20-RE-003) and the PRECISION Health Research, Singapore Clinical Implementation Pilot (PRECISE CIP) Fund. MH is supported by the JurongHealth Fund, PRECISION Health Research, Singapore Clinical Implementation Pilot (PRECISE CIP) Fund, the Breast Cancer Prevention Programme under Saw Swee Hock School of Public Health Programme of Research Seed Funding (SSHSPH-Res-Prog-BCPP), Breast Cancer Screening Prevention Programme under Yong Loo Lin School of Medicine (NUHSRO/2020/121/BCSPP/LOA), National Medical Research Council Clinician Scientist Award (Senior Investigator Category, NMRC/CSA-SI/0015/2017), the National University Cancer Institute Singapore (NCIS) Centre Grant Programme (CGAug16M005), NCIS Ng Teng Fong General Hospital Collaborative grant (CGAug16C002) and Asian Breast Cancer Research Fund. The funders had no role in study design, data collection and analysis, decision to publish, or preparation of the manuscript.

**Institutional Review Board Statement:** The study was conducted according to the guidelines of the Declaration of Helsinki, and approved by the Institutional Review Board of A*STAR Genome Institute of Singapore (Reference number: 2021-077, 6 May 2021) and Domain Specific Review Board by National Healthcare Group Singapore (Reference number: 2020/01327).

**Informed Consent Statement:** Informed consent was obtained from all subjects involved in the study.

**Data Availability Statement:** Data is contained within the article or supplementary material. The data presented in this study are available in Figures 1 and 2, Tables S2 and S3.

**Acknowledgments:** We wish to express our gratitude to the focus group and survey participants who shared their valuable insights and experiences. We also want to thank our dedicated research and administrative staff—Jenny Liu, Yen Shing Yeoh, Nur Khaliesah Binte Mohamed Riza, Ganga Devi D/O Chandrasegran, Hui Min Lau, Pooi Yee Wong, Hui Ling Tan, Kimiie Chia Wei Lin, Nabilah Binte Supiee, Nurfilya Binte Hamdil, Amanda Ong Tse Woon, Jing Jing Hong, Siew Li Tan, Evelyn Low Sok Peng, Marina Mohd, Noor Aisha Binte Mohamed Bahru Ali and Linus Chui for their contributions in the planning, preparation, and execution of BREATHE.

**Conflicts of Interest:** The authors declare no conflict of interest.

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
