# Peer review of "“It Will Lead You to Make Better Decisions about Your Health”—A Focus Group and Survey Study on Women’s Attitudes towards Risk-Based Breast Cancer Screening and Personalised Risk Assessments"

_curroncol, doi:10.3390/curroncol29120719_

Round 1

Reviewer 1 Report

Can the authors indicate in their methods whether the emotion-related survey question was inspired from previous work?

Good idea to choose to present the feeling distributions using figures. 

In the introduction, can the authors expand on what has been done in their country to promote mammography screening uptake? 

I suggest adding this reference to describe women's views related to risk-stratified breast screening: Mbuya Bienge, C., Pashayan, N., Brooks, J. D., Dorval, M., Chiquette, J., Eloy, L., . . . Nabi, H. (2021). Women's Views on Multifactorial Breast Cancer Risk Assessment and Risk-Stratified Screening: A Population-Based Survey from Four Provinces in Canada. Journal of Personalized Medicine, 11(2). https://doi.org/10.3390/jpm11020095 

Author Response

Dear Editor,

We appreciate the opportunity to submit a revised version of the manuscript. We thank the reviewers for their time and comments and have edited the manuscript to address their concerns. A version of the manuscript with changes tracked is uploaded together with this submission.

REVIEWER 1

  1. Can the authors indicate in their methods whether the emotion-related survey question was inspired from previous work?

Our response: We have added the clarification in bold below to the Methods section and attached a Word document of the survey as supplementary material.

“Participant responses were captured via a secure online questionnaire platform during the first visit, i.e. prior to the individual knowing their risk level [17]. Participants were required to state the extent to which they agreed with the statement “I am feeling <x> to receive my breast cancer risk report” on a 5-point scale (Strongly Agree, Agree, Neither Agree nor Disagree, Disagree, or Strongly Disagree). Eight feelings were assessed in total, including five negative (Scared, Regretful, Anxious, Worried, and Stressed Out), and three positive (Excited, Confident, and Optimistic). These feelings were included upon consensus of the BREATHE study team members in the design of the study recruitment experience survey (Supplementary File 1). Participants' ratings reflected their feelings in anticipation of receiving their risk reports in ~3 months.”

  1. Good idea to choose to present the feeling distributions using figures.

Our response: Thank you. Upon request of Reviewer 2, we have included additional barplots to represent the data in a different way.

Figure 3. Barplot representing number of responses indicating each positive emotion among 993 participants.

Figure 4. Barplot representing number of responses indicating each negative emotion among 993 participants

  1. In the introduction, can the authors expand on what has been done in their country to promote mammography screening uptake?

Our response: The nation-wide breast screening programme in Singapore is described in lines 50-64 below. We have included additional initiatives by the government to promote mammography screening (in bold).

“Singapore has one of the highest age-standardised breast cancer incidence rates in Asia [6]. With the intention of lowering breast cancer mortality in Singapore, the national breast cancer screening program (BreastScreen Singapore - BSS) was introduced in 2002 - the first in Asia [7]. Through tailored invitation letters and subsidies, the BSS program targets female Singaporeans and Permanent Residents 50 years of age and older for biennial screening. In addition, the local Health Promotion Board and various cancer societies hold activities and campaigns regularly to increase breast cancer awareness and promote the uptake of mammography screening. Women between the ages of 40 and 49 are encouraged to get screened but are urged to consult their doctor about the advantages and restrictions of screening mammography for their age group. However, even with a nation-wide screening programme, over 60% of Singaporean females between the ages of 50-69 years do not attend regular mammography at recommended intervals [7,8]. Studies over the past 25 years showed little change in women’s barriers and motivations to attend mammography screening, with recurring main themes like fear, cost, inconvenience, and a perceived lack of risk [9,10].”

  1. I suggest adding this reference to describe women's views related to risk-stratified breast screening: Mbuya Bienge, C., Pashayan, N., Brooks, J. D., Dorval, M., Chiquette, J., Eloy, L., . . . Nabi, H. (2021). Women's Views on Multifactorial Breast Cancer Risk Assessment and Risk-Stratified Screening: A Population-Based Survey from Four Provinces in Canada. Journal of Personalized Medicine, 11(2). https://doi.org/10.3390/jpm11020095

Our response: This reference is included in lines 570-572:

“An online survey by Bienge et al on 4293 women aged 30 to 69 years from Canada found 63.5% to 72.8% of them to be favourable towards risk-stratified breast cancer screening [47].”

Reviewer 2 Report

In this manuscript, the authors surveyed 993 participants in a risk-based mammography screening (MAM) study on how they felt in anticipation of receiving their risk profiles. Focus group discussions (FGD) were conducted with 54 women with no BC history. The results showed that attitudes towards mammography screening have remained unchanged for over 25 years. Additionally, they found that FGD participants reported that they would be more likely to attend routine mammography after getting their BC risks assessed, despite uncertainty and concerns about risk-based screening. Their results support the empowering of Singaporean women with personal health information to improve MAM uptake. The results are significant for breast cancer screening and precision prevention approaches; however, the following issues are required for explaining:

1. Pie chart or bar chart should be used instead of Venn chart in displaying the data of a survey. Please add the above two charts.

2. Some other characteristics of participants should be provided, for example, family history of breast cancer/other cancer.

3. The authors are recommended to provide a word version of the survey as a supplementary file to clearly display the items.

4. Although the manuscript is understandable, there are still some grammatical and spelling errors throughout which could be easily corrected.

Author Response

Dear Editor,

We appreciate the opportunity to submit a revised version of the manuscript. We thank the reviewers for their time and comments and have edited the manuscript to address their concerns. A version of the manuscript with changes tracked is uploaded together with this submission.

REVIEWER 2

In this manuscript, the authors surveyed 993 participants in a risk-based mammography screening (MAM) study on how they felt in anticipation of receiving their risk profiles. Focus group discussions (FGD) were conducted with 54 women with no BC history. The results showed that attitudes towards mammography screening have remained unchanged for over 25 years. Additionally, they found that FGD participants reported that they would be more likely to attend routine mammography after getting their BC risks assessed, despite uncertainty and concerns about risk-based screening. Their results support the empowering of Singaporean women with personal health information to improve MAM uptake. The results are significant for breast cancer screening and precision prevention approaches; however, the following issues are required for explaining: 

  1. Pie chart or bar chart should be used instead of Venn chart in displaying the data of a survey. Please add the above two charts.

Our response: Bar charts for positive (Figure 3) and negative emotions (Figure 4) have been added.

Figure 3. Barplot representing number of responses indicating each positive emotion among 993 participants.

Figure 4. Barplot representing number of responses indicating each negative emotion among 993 participants.

  1. Some other characteristics of participants should be provided, for example, family history of breast cancer/other cancer.

Our response: We have described several participant characteristics in Table 2, including family history of breast cancer and screening history.

Table 2: Characteristics of 993 survey participants.

Characteristics

Number (%)

Age (median)

49 years

Ethnicity

            Chinese

764 (77%)

            Malay

119 (12%)

            Indian

60 (6%)

            Others

50 (5%)

Family history of breast cancer

108 (11%)

            Mother

50 (5%)

            Sister 

60 (6%)

            Daughter 

0 (0%)

Ever attended breast cancer screening

            Yes

755 (76%)

            No

238 (24%)

  1. The authors are recommended to provide a word version of the survey as a supplementary file to clearly display the items.

Our response: The recruitment experience survey has been added as Supplementary File 1:

“Participant responses were captured via a secure online questionnaire platform during the first visit, i.e. prior to the individual knowing their risk level [17]. Participants were required to state the extent to which they agreed with the statement “I am feeling <x> to receive my breast cancer risk report” on a 5-point scale (Strongly Agree, Agree, Neither Agree nor Disagree, Disagree, or Strongly Disagree). Eight feelings were assessed in total, including five negative (Scared, Regretful, Anxious, Worried, and Stressed Out), and three positive (Excited, Confident, and Optimistic). These feelings were included upon consensus of the BREATHE study team members in the design of the study recruitment experience survey (Supplementary File 1). Participants’ ratings reflected their feelings in anticipation of receiving their risk reports in ~3 months.”

  1. Although the manuscript is understandable, there are still some grammatical and spelling errors throughout which could be easily corrected.

Our response: Thank you. We have reviewed the grammar and spelling errors using language assistant software.

Round 2

Reviewer 2 Report

The revised manuscript has made a great improvement. I have no more comments and recommends.